# Gut Microbiota Ecosystem Governance of Host Inflammation, Mitochondrial Respiration and Skeletal Homeostasis

**DOI:** 10.3390/biomedicines10040860

**Published:** 2022-04-06

**Authors:** Wei-Shiung Lian, Feng-Sheng Wang, Yu-Shan Chen, Ming-Hsien Tsai, How-Ran Chao, Holger Jahr, Re-Wen Wu, Jih-Yang Ko

**Affiliations:** 1Core Laboratory for Phenomics and Diagnostics, Department of Medical Research and Chang Gung University College of Medicine, Kaohsiung Chang Gung Memorial Hospital, Kaohsiung 83301, Taiwan; lianws@gmail.com (W.-S.L.); wangfs@ms33.hinet.net (F.-S.W.); ggyy58720240@gmail.com (Y.-S.C.); 2Center for Mitochondrial Research and Medicine, Kaohsiung Chang Gung Memorial Hospital, Kaohsiung 83301, Taiwan; 3Department of Child Care, College of Humanities and Social Sciences, National Pingtung University of Science and Technology, No.1, Shuefu Road, Pingtung 91201, Taiwan; alantsai@mail.npust.edu.tw; 4Emerging Compounds Research Center, General Research Service Center, National Pingtung University of Science and Technology, No.1, Shuefu Road, Pingtung 91201, Taiwan; hrchao@mail.npust.edu.tw; 5Department of Environmental Science and Engineering, College of Engineering, National Pingtung University of Science and Technology, No.1, Shuefu Road, Pingtung 91201, Taiwan; 6Department of Anatomy and Cell Biology, University Hospital RWTH, 52074 Aachen, Germany; hjahr@ukaachen.de; 7Department of Orthopedic Surgery, Maastricht University Medical Center, 6229 ER Maastricht, The Netherlands; 8Department of Orthopedic Surgery, College of Medicine, Chang Gung University, Kaohsiung Chang Gung Memorial Hospital, Kaohsiung 83301, Taiwan; ray4595@cgmh.org.tw

**Keywords:** oxidative stress, aging, microbiota, gut dysbiosis, mitochondrial, bone and joint disease

## Abstract

Osteoporosis and osteoarthritis account for the leading causes of musculoskeletal dysfunction in older adults. Senescent chondrocyte overburden, inflammation, oxidative stress, subcellular organelle dysfunction, and genomic instability are prominent features of these age-mediated skeletal diseases. Age-related intestinal disorders and gut dysbiosis contribute to host tissue inflammation and oxidative stress by affecting host immune responses and cell metabolism. Dysregulation of gut microflora correlates with development of osteoarthritis and osteoporosis in humans and rodents. Intestinal microorganisms produce metabolites, including short-chain fatty acids, bile acids, trimethylamine N-oxide, and liposaccharides, affecting mitochondrial function, metabolism, biogenesis, autophagy, and redox reactions in chondrocytes and bone cells to regulate joint and bone tissue homeostasis. Modulating the abundance of *Lactobacillus* and *Bifidobacterium*, or the ratio of Firmicutes and Bacteroidetes, in the gut microenvironment by probiotics or fecal microbiota transplantation is advantageous to suppress age-induced chronic inflammation and oxidative damage in musculoskeletal tissue. Supplementation with gut microbiota-derived metabolites potentially slows down development of osteoarthritis and osteoporosis. This review provides latest molecular and cellular insights into the biological significance of gut microorganisms and primary and secondary metabolites important to cartilage and bone integrity. It further highlights treatment options with probiotics or metabolites for modulating the progression of these two common skeletal disorders.

## 1. Introduction

Osteoporosis (OP), osteoarthritis (OA), and sarcopenia are significant causes of musculoskeletal disability in the elderly in developed countries. These chronic diseases are life-threatening and overwhelm patients’ workforce, life quality, and mental health. The development of these diseases is irreversible and correlates with genetic background, genomic instability, cellular senescence, and stem cells plasticity loss [1,2]. Increasing studies have revealed intestinal microflora composition in the development of age-association tissue deterioration and tumorigenesis [3,4]. Intestinal microorganisms involve non-communicable diseases, including metabolic disorders, neurodegenerative diseases, cardiomyopathy, cancers, autoimmune diseases, and musculoskeletal disorders [1,5]. Of the degenerative diseases, the management of sarcopenia, OA, and OP in senile patients have become a tremendous socioeconomic drain and healthcare burden [6,7]. How gut dysbiosis affects host bone and cartilage integrity and accelerates senile OP and OA development is the topic of growing discussion.

The mammalian gastrointestinal tract microenvironment is a sophisticated ecosystem harmonized by commensal microbiota and the host’s symbiotic reactions. There is increasing evidence that the gut microenvironment is widely distributed and has a biodiversity of microorganisms such as microbiome, commensal, symbiotic, and pathogenic microorganisms. The composition of gut microorganisms plays a vital role in host tissue integrity and correlates with age-induced tissue deterioration [8,9]. Intestinal microorganism alteration correlates with inflammation, oxidative stress, anabolism repression, dysregulating innate immunity, and energy metabolism. The biological contribution of gut microorganisms to bone and joint tissue homeostasis in physiological and pathological conditions warrants a broad spectrum of insight.

This review aims to deliver new insight into the contribution of gut microorganisms to the development of OA and OP in senile population and the biological roles of microorganism species and primary or secondary metabolites response to redox and inflammatory reactions in host bone and joint microenvironments, as well as illustrate the remedial potential of gut microbiota and metabolite for preventing OA and OP development.

## 2. Multifaceted Functions of Gut Microorganism to Host Tissue Integrity

Expanding studies have shown that gut microorganisms affect a plethora of host tissue or organ function, including brain function [10], hepatic microenvironment [11], and joint integrity [12]. Table 1 reveals the crosstalk between the biological role of metabolites and gut microorganisms associate vascular and endothelium metabolism [13,14], acute pancreatitis [15,16], intestinal homeostasis [17], skin aging [18], myeloid cells-dependent autoimmune disorders [19], lung infection and Covid19-induced lung damage [20,21], cardiac inflammation [22], Parkinson disease, Alzheimer disease, and autism [23,24,25,26], hepatic steatosis [27,28], sarcopenia [29,30], diabetes-induced OP [31,32], male [33,34] and female [35,36] reproduction functions and disease. These studies also single out certain microorganisms, which influence cellular activities through regulating inosine-A2AR/peroxisome proliferator-activated receptor gamma (PPARγ), chemokine, antioxidant protein nuclear factor, Nrf2 (erythroid-derived 2), NF-κB (nuclear factor-κB), and GABA (gamma-aminobutyric acid) signaling pathways. These investigations hint that gut microorganisms directly or indirectly affect host cellular bioenergy activities and the homeostasis function of tissues.

The harmonization of billions of microbial taxa in the intestinal tract microenvironment promotes the interaction of bacteria and intestinal tract epithelium to stabilize immune response. On the other hand, gut microorganism increases the production of metabolites, which are advantageous to host health [37]. The coordination of billions of microbial taxa in the gut microenvironment maintains bacterial interactions of gut epithelial cells that promote the production of dietary biotransformation metabolites by gut microbes to stabilize host immune activity and benefit the host physiological responses. For example, nutrient-balanced diets enhance gut microorganism accommodation and metabolite production, upregulating host energy utilization, immune response, and cellular metabolism [38,39]. Gut microbes produce bioactive molecules into the bloodstream, including LPS (liposaccharides), glycoproteins, SCFAs (short-chain fatty acids), phospholipids, and vitamins, which are implicated in cellular activity and metabolism in host tissue. Gut leaky or dysbiosis increases unwanted molecules, aggravating pro-inflammatory cytokines to flare up inflammation and oxidative radicals [40]. In addition, gut microorganism-derived metabolites affect transcriptomic landscapes, epigenetic landmarks, and enzyme activities, modulating mitochondrial biogenesis, metabolism, and redox reaction in the host cellular microenvironment [41,42].

Maintaining gut biodiversity and interacting with host intestinal epithelium has become an emerging strategy to keep the host healthy. Probiotics have been utilized to sustain microbes’ proliferation, diversity, and metabolism, further promoting the interplay of host gut epithelium [43,44]. In regular conditioning, gastrointestinal bacteria build a broad range of biologically active molecules, such as metabolite, protein, and enzymatic. In this sense, transplantation of fecal microbiota from a healthy donor is found to reverse the ecosystem in the recipient animals who contract OA or OP [45,46,47]. These strategies highlight the therapeutic potential of gut microorganism intervention for improving chronic bone and joint diseases. On the other hand, gut microbiota metabolizes proteins, lipids, carbohydrates from diets into a plethora of small molecule metabolites, including SCFAs, secondary bile acid, and TMAO (trimethylamine N-oxide) and have remedial effects on host metabolism [48].

## 3. Cellular and Molecular Events Underlying Osteoarthritis (OA) Development

OA is a common chronic musculoskeletal disorder that is highly prevalent in senior people. The OA condition is heterogeneous and irreversible with many signs, including swelling, pain, mobility repression, and deformity in the injured joints, which badly affects patients’ daily activity. This joint disorder has become a massive drain of healthcare resources [49]. This disease is frequently present in the weight-bearing side of the knee, hip, and spine in young and aged patients. While the cause of OA remains uncertain, the severity of this disease correlates with patients’ age, gender, genetic background, body mass index, and history of mechanical disuse [48].

### 3.1. Molecular Events in Aged Cartilage Microenvironment

Age-mediated OA is a chronic degenerative process with low-grade inflammation in the injured joints. Forty percent of the patients over 65 years old have a history of this joint disorder, which has become an essential medical care issue around the globe [50]. Chondrocytes play a critical role in producing abundant extracellular matrices to form well-structured articular cartilage. Chondrocyte senescence is one of the prominent cellular events, accelerating chondrocyte loss, extracellular matrix underproduction, and the production of proteolytic enzymes, which progressively accelerate cartilage degradation in the OA joint [51]. The initiation pathogenesis of OA is multiple factors accompanied by an intricate network of molecules associated with innate immune activity. Chondrocytes promote the inflammatory process in OA by upregulating the expression of TLR2 and TLR4 (Toll-like receptor), directly or indirectly activating MMPs (matrix metalloproteinases) such as MMP-1, MMP-9, and MMP-13, which contribute to cartilage deterioration [51,52].

These harmful circumstances in the OA microenvironment also induce epigenetic alteration, which disrupts gene transcription related to cartilage anabolism, including aggrecan, collagens, NFAT1 (nuclear factor of activated T cells), and Sox9 (sex-determining region Y box 9). Furthermore, it upregulates catabolic processes by increasing aggrecanases, collagenases, inflammatory cytokines, and Runx2 (runt-related transcription factor 2) that increase ECM (extracellular matrix) proteolysis or calcified matrix production in chondrocytes the development of cartilage erosion and osteophyte formation around the injured joints [53]. ECM is a highly abundant and critical component of tissues that dominates the architectural integrity of cellular metabolism such as anabolic and catabolism [54]. Emerging results also suggest that the ECM is responsible for counteracting dysbiosis-induced pathogenic bacteria, metabolite imbalances, and attracting immune cell invasion [55].

### 3.2. Endotoxin and Low-Grade Inflammation

A study by Lorenzo et al. has highlighted the association of *Porphyromonas gingivalis*, intestinal *Prevotella copri*, and the pathophysiology of human RA (rheumatoid arthritis) and OA [56]. It is known that the fluctuating changes of gut microbiota compositions depend on host lifestyle, living surroundings, and physiological status. Age is crucial in redistributing intestinal microorganisms and becomes a risk factor for human inflammatory joint disorders, hinting that sophisticated reciprocal actions are present in intestinal microbes and host injured joints. Several studies showed that all Gram-negative bacterial constitutions assembled and secreted vesicles. The ability of vesicles to have diverse contents and insensitivity to proteases allows vesicles to enter the host circulatory system to trigger biological activity in distant tissues [57,58]. The Gram-negative bacteria produce LPS, a soluble microorganism cell wall component, and endotoxin factor. The level of LPS increases entry into the host bloodstream, and synovial fluid triggers a pro-inflammatory response, synovitis, and articular cartilage destruction in the development of arthritis [59]. Immune activity by LPS in chondrocytes leads to an induced cascade of complement factors secretion and activation, such as complement C3 and factor B, mimecan (osteoglycin), PTX3 (pentraxin 3), and various cytokines production [52]. However, little is known about the contribution of specific bacterial composition to cartilage metabolism, pain, and severity during age-mediated OA and inflammatory joint diseases [60,61].

### 3.3. Oxidative Stress and Inflammation in Chondrocyte Senescence

OA development is considered a low-grade inflammation with a long process in articular cartilage and synovial compartments by age-induced joint degeneration. In the inflammatory condition, mitochondrial dysfunction and ROS (reactive oxygen species) production consequently increased SASPs (senescence-associated secretory phenotypes) in chondrocytes to accelerate cellular senescence and overburden in osteoarthritic cartilage [62]. Aging chondrocytes show senescence associate markers, including p53, p16^INK4a^ (cyclin-dependent kinase inhibitor 2A), p21^CIP1^ (cyclin-dependent kinase inhibitor 1A), and ROS overproduction together with mitochondrial dysfunction, ATP underproduction, and SA-β-gal (senescence-associated β-galactosidase) overactivation. In addition, senescent chondrocytes secrete a plethora of soluble factors, such as IL-6 (Interleukin-6), IL-17, IL-1β, MMPs, TGF (transforming growth factor), and TNFs (tumor necrosis factors), which are found to deteriorate chondrocyte function and cartilage integrity [63,64].

In the IL-1β-stressed chondrocytes as an in vitro model of OA development, mitochondrial dysfunction and energetics loss increase NO (nitric oxide) production together with increased ILR1 and TNFRII signaling in the chondrocytes [65,66]. IL-1β in the joint microenvironments also attracts plenty of immune cells, including macrophages, neutrophils, and lymphocytes, upregulating adipocyte formation in the articular compartment [67]. Although the extracellular matrix type II collagen and proteoglycans are essential components of chondrocytes and articular cartilage structure, the biosynthesis of these matrices is reduced in the early stages of OA and is associated with mitochondrial dysfunction, oxidative stress, and gut dysbiosis related [68]. Cartilage metabolism and chondrocyte activity are mainly regulated by the respiratory chain reaction in the mitochondrial compartment. AMPK (AMP-activated protein kinase) and mTOR (mechanistic target of rapamycin) pathways involve oxidative phosphorylation pathway, glucose uptake, glycolysis, and ROS production [69,70]. Accumulating studies have revealed the importance of bioenergetic programs of mitochondrial organelle in articular cartilage homeostasis, chondrocyte function, and OA development [71,72]. The emerging insights into the interaction of microbiota-derived metabolites and mitochondrial activity in chondrocytes during joint degeneration may underpin the rationale for developing new treatment options for OA through modulating mitochondrial metabolism.

## 4. Gut Microbiome and Host Articular Cartilage Homeostasis

Increasing shreds of evidence have shown the involvement of gut microbiota in host biological responses during the aging program. An interplay of microbiota and host tissue homeostasis is that microbiota affects the gut ecosystem and produces metabolites, which regulate tissue immunity. In the studies of major histocompatibility complex, class II, DR beta 1 (HLA-DRB1) 0401 and 0402 mice, Gomez et al. [73] reveal that the gut microflora composition and gut permeability depend on host genetic background and age. These data also shed light on the gut-joint axis crosstalk and single out the abundance of specific bacteria, including Allobaculum sp., Barnesiella sp., and Parabacteroides sp., which correlates with the number of particular bacteria the development of OA, RA, and immune activities.

Furthermore, specific probiotics like Bifidobacterium and Lactobacillus have demonstrated perturbation immune checkpoint activity and immune response in the host. Enrichment of *Lactobacillus rhamnosum* and *Bifidobacterium breve* administrated in mice showed that microorganisms ameliorated inflammatory cytokines, block CTLA-4 (cytotoxic T lymphocyte-associated protein 4), stimulated host immune system response to anti-PD-L1, and optimized Tregs (regulatory T cells) functions [74]. RA is a long-term autoimmune disorder, resulting in chronic inflammation and accompanying bone and cartilage injury in joints [75]. Gut dysbiosis increases pathogenic microorganisms that are potent inducers of proinflammatory Th17 (T helper 17) and implicated in the inflammatory, autoimmune diseases and substantially influences immune responses between host microbiota and rheumatoid [76].

Collective analysis of clinical specimens has revealed that gut microbiota composition and its metabolites are linked to various types of tissue dysfunction. Gut dysbiosis, a state of commensal disturbance in the gut ecosystem, disrupts microbiota and gut microenvironment communications, disturbing host immune reaction to aggravate obesity, bowel syndrome, RA, and musculoskeletal disorders [77].

### 4.1. The Correlation of Gut Microbiota and Human and Rodent OA

The investigations of a large cohort study conducted by Boer et al. have shown that the burden of gastrointestinal disorders induces endotoxin overproduction, which upregulates low-grade inflammatory reaction and joint pain in knee OA microenvironment [61]. The abundance of *Streptococcus species* (spp.) strongly correlates with the severity of articular cartilage inflammation and is considered one of OA’s risk factors. The study also shows the abundance of *Clostridium* spp., increases in the OA group, whereas the enrichment of *Bifidobacterium longum* and *Faecalibacterium prausnitzii* is significantly decreased [78]. In addition, patients from different ethnic groups who are overweight with or without OA signs have different gut microbiota profiles, including *Gemmiger*, *Klebsiella*, *Akkermansia*, *Bacteroides*, *Prevotella*, *Alistipes* and *Parabacteroides* [79]. These investigations suggest that the condition of knee OA is linked to gut microbiota alteration (Figure 1).

Further explanation of gut-joint axis interaction, antibiotic-induced gut dysbiosis, reduced serum of LPS, and the inflammatory response is required. These lead to a decrease in TNF-α, IL-6, and MMP-13 expression and impairment of pathogenesis of OA [71]. The analysis of gut dysbiosis may, at least in part, explain the systemic point of view between joint inflammation, articular cartilage, and gut flora during age-mediated OA development.

In germ-free (GF) and specific pathogen-free (SPF) animal models, the analysis of proof of concept has shown that unbalanced diet mediated metabolic syndromes, including obesity and diabetes, accelerates gut microbiota dysbiosis together with the development of OA. With regard to microflora, mice with knee OA have a significant loss in *Bifidobacterium pseudolongum* together with significant increases in *Peptostreptococcaceae* sp., *Peptococcaceae rc4-4* sp., and *Peptococcaceae* spp. [80]. Ullici et al. also revealed concrete evidence regarding the contribution of gut microbiota in SPF mice to the severity of OA signs in DMM (destabilized medial meniscus)-mediated knee joint injury compared to GF animals. Twenty-eight relative OTU (operational taxonomic units) are affected and contribute to knee OA development [81]. However, little is known about which metabolites are produced by these specified gut microorganisms and whether these molecules influence joint inflammation or OA development. There is still a long way to go to have productive insight into the dynamic process of host and gut microbiome and the culturomic landscapes of specified gut microorganisms to understand how gut microbiota changes chondrocyte function and joint homeostasis.

### 4.2. Aging Contributes to Intestinal Dysbiosis and Hosts Mitochondrial Function and Metabolic Dysregulation

Age-mediated metabolic dysregulation is characterized by low incorporation of nutrients, genetic instability, decreased intracellular communication, and gut microflora alteration [82]. Host genetic background affects gut microbiome landscapes and the ratio of Firmicutes and Bacteroidetes, which play an essential role in the host metabolism. For example, the gut microflora profile correlates with host ethnicity, and HD5 (human defensin 5), one of Paneth cell a-defensins, regulates human gut microorganisms. The elderly people have fewer HD5 levels together with significant reductions in *Blautia*, *Anaerostipes*, and *Fusicatenibacter* et al., compared to the middle-aged people [83]. The investigations of transgenic or knockout mice fed with different diets or high-fat diets also underpin the importance of host genetic background in the interplay of host and gut microbiota during cartilage erosion. For example, TLR (Toll-like receptor) 5 knockout mice develop metabolic syndromes with body overweight and hyperglycemia together with knee OA signs upon high-fat diet consumption [82,84]. Age and abnormal diet risk factors caused low-grade inflammation associated with LPS activity in OA, which involves the macrophage, neutrophil response, and activation of TLR pathways. Upregulation of TLR2, 4, and 5 is responsible for the onset of early OA, triggering the innate immune system. Notably, LPS excites significantly shaped gut symbionts and reduces the number of the Gram-positive *Bifidobacterium* spp. [85]. Mice deficient in TLR2 show a significant reduction in autoimmunity and vital tight junction-associated protein ZO-1(zonula occludens-1) signaling, together with high gut permeability and bacterial translocation [86]. Gnotobiotic or antibiotics-treated mice have fewer OA signs with less inflammation than conventional mice [81,82].

Several studies noted a correlation between microbiota quality, diversity, metabolites, and mitochondrial activity. Clinical research implicated mitochondria–microbiota inter-talk that regulate host immune response and energetic metabolism [87]. Pathogen direct metabolites such as LPS, flagellin, lipoteichoic acid, lipoproteins, and PAMPs (pathogen-associated molecular patterns) are targeted to affect the host cellular processing of glycolytic, mitochondrial fatty acid oxidation, and ATP production. Interestingly, mitochondrial ROS may also regulate the intestinal epithelial barrier, cell motility, epithelial cell proliferation, and immunosuppression. Notably, reciprocal interactions between microflora and host mitochondrial are associated with specific microflora, such as *Mycobacterium tuberculosis* downregulates the LPS-mediated inflammation pathway, and *Ehrlichia chaffeensis* increased MnSOD activity [88,89].

### 4.3. Gut Microbiota Changes Host Cellular Energy Metabolism

Gut microbiota affects subcellular organelle function in chondrocytes during OA joint development. It modulates mitochondrial function regulators AMPK activity and PGC-1α (peroxisome proliferator-activated receptor-gamma coactivator-1 alpha) signaling to control autophagy and oxidative stress and inflammation. Gut microorganisms also influence energy metabolism, ECM anabolism, and mitochondrial microenvironment integrity [90]. Gut dysbiosis interrupts nutrient incorporation and compromises SCFAs and metabolite production by Bacteroidetes, which repress articular cartilage tissue homeostasis together with decreased AMPK and PGC-1α pathways accelerate synovitis and OA development [91].

Interruption of glucose metabolism in mice by administration with D-galactose and beryllium salts increases oxidative stress, cellular inflammation, and gut dysbiosis, upregulating age-mediated tissue degeneration, including cognitive decline [92]. Supplement with Cistanche polysaccharides is found to maintain host-gut microbial homeostasis, improving innate immunity, gut microbiota composition, and amino acid metabolism. Systemic inhibition of glycolysis in chondrocytes represses mitochondrial function and induces OA signs. Supplement with galactose represses glycolysis and promotes the enzyme activity for OXOPHO (oxidative phosphorylation) to increase ATP production in chondrocytes [93]. Galactose appears to maintain mitochondrial energetics, which increases mitochondrial respiration and the TCA cycle to compromise OA development through activating the AMPK-PGC-1α pathway.

## 5. Reciprocal Control of Mitochondrial Redox and Inflammation in Gut-Joint Axis

Increasing evidence has revealed that the gut-joint pathway correlates with knee OA development. A multi-cohort study with >2500 participants shows that commensal microbes are essential to many diseases, including knee OA [6]. Feeding diets containing probiotics reduced the expression of PTGS2 (prostaglandin-endoperoxide synthase 2) and TGF-β1 (transforming growth factor-β1) that promote inflammatory signaling. Probiotics regulated collagen II expressions, improved ECM synthesis in articular cartilage, and repressed OA development [94]. Furthermore, the multistrain probiotic composition is responsible for strengthening the chondroitin sulfate activity and attenuating OA progress [95]. Probiotics mainly with Firmicutes are found to metabolize dietary fibers and nondigestible starch in the host gut to produce SCFAs as an additional energy source for regulating energy expenditure, which is advantageous to host body weight control [96].

### 5.1. Probiotics and Short-Chain Fatty Acids (SCFAs)

*Lactobacillus* and *Bifidobacterium* are important bacteria and are widely used as probiotics, maintaining the gut microenvironment homeostasis to protect from unwanted microorganisms by producing lactic acids. Of the taxa, *B. longum*, *L. casei*, *L. Shirota,* and *L. acidophilus* have become probiotic options for slowing OA development by repressing inflammation and reversing cartilage anabolism [78,97]. In addition, the abundance of *Clostridiales*, *Roseburia*, *Lachnospiraceae,* and *Erysipelotrichaceae* are utilized to promote SCFAs production for protecting the digestive tract from inflammation and ROS damage, as well as enhance the flavin/thiol electron transport to preserve microbial diversity in the gut microenvironment [98].

Probiotics and SCFAs are advantageous to modulate Firmicutes-to-Bacteroidetes ratio to maintain gastrointestinal microenvironment homeostasis. Of the SCFAs, n-butyrate, acetate, and propionate are essential sources of carbon influx of gut microorganisms from the diet. These fatty acids can circulate in the bloodstream to directly or indirectly affect metabolism, physiological activity, and pathological reactions in host tissue [6]. Probiotics-derived SCFAs are found to control oxidative stress directly and inflammation, improving metabolism or energy consumption in host tissue [99].

### 5.2. Short-Chain Fatty Acid Control of Host Intracellular Signaling

Increasing studies have uncovered that GPR-41 (G protein-coupled receptors-41) and GPR43 are SCFAs-responsive molecules, which communicate gut microbiota-derived signaling into the host intracellular microenvironment. These two receptors regulate plenty of physiological activities through controlling epigenetic landscapes. For example, butyrate upregulates H3K9ac (histone H3K9 acetylation), promoting PPARγ/CD36/STAR (steroidogenic acute regulatory) protein pathways and activates PGC1-α signaling to maintain mitochondrial integrity and repress oxidative stress in ovarian granulosa cells [100]. Administration with sodium butyrate induces GRP43, downregulating inflammation and chemotaxis production in chondrocytes from osteoarthritic cartilage [101]. Zhou et al. has revealed that sodium butyrate treatment attenuates inflammation and ROS generation, reversing extracellular matrix degradation in inflamed chondrocytes. These effects are advantageous to slow down the development through regulating PI3K (phosphoinositide 3-kinase)/Akt (protein kinase B)/mTOR (mammalian target of rapamycin) pathway-mediated autophagic programs in chondrocytes [102]. Investigations of in vitro and in vivo models have suggested that probiotics or SCFAs are provided with chondroprotective effects on experimental OA development. Whether these agents can slow human OA warrants double-blind clinical trials.

### 5.3. Fecal Microbial Transplantation (FMT) as a Probiotics Option

Given the modulation of gut microbiota by probiotics, moderate amounts of antibiotics, or metabolites of gut bacteria are beneficial for suppressing inflammation, increasing cellular metabolism, and musculoskeletal tissue homeostasis [94,103,104], transplantation from healthy donor fecal microbiota may be a therapeutic option. For example, oral administration with probiotics *Lactobacillus casei* together with collagen II and glucosamine significantly represses pro-inflammatory cytokine production and promotes IL-4 and IL-10 expression. *L. casei* appears to have the potential to reduce oxidative stress, synovitis, and immune response in OA knee joints in rats [104]. Administration with *L. rhamnosus*, *L. acidophilus* and *L. paraca* is found to ameliorate articular cartilage degradation, pain, and inflammation and control cartilage catabolic factors through downregulating TNF-a signaling and pro-inflammatory cytokines [105]. FMT (fecal microbiota transplantation) combined with antibiotics treatment is found to attenuate loss in knee joint morphology during OA development. Huang et al. [106] transplants feces from mice with knee OA conditions into recipient GF mice. The knee joints in the recipients show endotoxemia together with severe OA signs, including inflammation and cartilage loss in meniscal ligamentous injury-induced knee joints. Analysis of 16S rRNA and RNA sequencing also uncover significant increases in the abundances of *Fusobacterium*, *Faecalibaterium*, and *Ruminococcaceae*, which correlate with increased inflammatory cytokines in severe OA joints.

### 5.4. The Limitations of Gut Microbiota for Controlling OA Development

While accumulating evidence has highlighted the biological roles of gut microbiota in host knee joint integrity, gut microbiota profile is dynamically changed depending on host gender, age, disease types and severity, lifestyle, diet, etc., [107]. Repression of gut dysbiosis by probiotics, diets, or exercise can restore gut microflora diversity and change knee joint inflammation, pain, and degeneration. Whether probiotics or microbiota-derived metabolites have remedial effects on human OA joint warrants investigations. The FMT has been found to improve human autoimmune disease; however, given the multifaceted regulatory mechanisms of immune responses, this alone may not infer therapeutic efficacy in OA. Nonetheless, the development of OA appears to be a multifactorial condition, including biomechanical and kinematic disorders. Modulating gut microbiota and its metabolites by diet or FMT to prevent OA development may require more specific proof-of-concept evidence and double-blind clinical trials. Further, we enumerate the results of recent clinical trials related to FMT in various diseases, demonstrating its potential for future clinical applications (Table 2).

## 6. Gut Microbiota Modulation of Osteoporosis (OP)

In addition to OA, OP is considered one of the life-threatening musculoskeletal diseases in the elderly. This common bone disorder is characterized by poor bone quality, including low bone mass, sparse microstructure, and decreased biomechanics. Menopause and age are important etiological factors of this bone disease, putting patients at risk of osteoporotic fracture. The development of senile OP is progressive and irreversible and ultimately devastates patients’ kinematical activities, life quality, and even survival [113]. While the bone quality is found to decline with patients’ age, slowing down the development of OP by physical exercise, sufficient vitamin D intake, and diets with abundant calcium and nutrients is widely suggested [114]. Osteocytes play an essential role in keeping bone tissue in a net gain state. They produce a mineralized extracellular matrix and interact with >100 proteins to form the mineralized network in the skeleton. Osteoclast overactivation induces excessive bone resorption [115] and marrow fat overproduction [116], which are also prominent features of the osteoporotic disease. A plethora of age-induced deleterious reactions, including oxidative radical overproduction [117], mitochondrial dysregulation [118], and osteogenic progenitor cell dysfunction [119], accelerates the loss in bone formation capacity. Because diets influence bone quality, gut microbiota seems important to bone mass homeostasis.

### 6.1. Age-Related Osteoporosis and Bone Cell Senescence

Host degenerative programs increased cell senescence in which metabolic activities rather than division capacity are still working in the cell microenvironment. Cellular senescence is a notable feature of age-induced tissue deterioration [120,121]. Once cells shift into senescence, a plethora of molecular events, including the p16^Ink4a^, p21^Cip1^, and p53 signaling, mitochondrial dysfunction, oxidative stress, cell growth arrest, b-galactosidase activation, and the production of soluble cytokines or bio-active molecules (senescence-associated secretory phenotypes, SASP) are present in senescent cells [121]. Expanding reports have shown that senescent osteoblast (osteocyte) overburden is present in age-induced osteoporotic skeleton [122] and damaged bone tissue upon γ-ray irradiation [123]. Modulating senescence-associated secretory phenotypes by genetic or pharmaceutical tools can reverse osteogenic differentiation of osteogenic precursor cells [124] and preserve bone mass and microstructure integrity in aged mice [125]. The loss in autophagy, an organelle for disposal of macromolecules in the unutilized organelle, accelerates senescence of osteogenic precursor cells, upregulating loss in bone mineral density of old mice [126]. In addition, plenty of intracellular mechanisms are found to counteract the senescence program during muscle dystrophy, hormone deficiency, and joint injury-mediated bone loss. As the bone tissue homeostasis in the osteoporotic skeleton is disrupted by the imbalanced bone-making cell and bone-resorbing cell activities, current pharmaceutical intervention strategies center on promoting bone formation or repressing excessive bone turnover. Recent investigations have revealed the association of gut microbiota and functional metabolites with host bone metabolism and immune activity, which exhibited to affect osteoimmunity [127,128].

### 6.2. Osteoimmunity in Bone Mass Homeostasis and Osteoporosis

Menopause-induced estrogen deficiency increases the risk of metabolic syndromes, excessive osteoclastic resorption, and osteoporotic fracture, especially the aged postmenopausal females with osteoporotic fracture have a high risk of complicated morbidity and preterm mortality. Dysregulated bone-making cells mediated bone formation and osteoclasts mediated remodeling aggravate bone mass loss and microstructure deterioration in osteoporotic bone. Mesenchymal stem cells are the primary source of osteoblasts and osteocytes, which work on mineralized matrix production and network construction. On the other hand, macrophages in peripheral blood or bone marrow shift to osteoclasts, which produce protons and proteinases to remodel the mineralized network. Estrogen loss induces chronic inflammation or oxidative stress, upregulating osteoclast formation or changing osteoblasts coupling with osteoclasts. The cytokine RANKL (receptor activator of nuclear factor κB) is indispensable in osteoclast formation, activation, and survival. In addition, TGF-β1 (transforming growth factor-beta1), IL-4, IL-10, Th17, and Treg cells are also found to regulate osteoclast activity and remodeling capacity in the bone microenvironment [129,130].

Ohlsson et al. [131] have interpreted “osteomicrobiology”, which includes bone physiology, immunology, and microbiology, and shed light on the importance of gut microbiota in bone health and OP development. Increasing evidence has revealed that gut dysbiosis or gut microflora-mediated disorders (the change of symbiotic gut microbes) correlate with senile OP in humans and rodents. Host metabolism status is linked to gut microflora profiles [3]. Microorganisms are discovered to activate host immune response through contacting intestinal epithelial cells. Microbiota-derived metabolites circulate into blood flow toward host tissues and communicate between tissues or cells, thus affecting signaling transmission [132]. Germ-free mice show phenotypes of higher bone mass together with decreased osteoclast formation and low inflammatory cytokines, including TNF-α and IL-6 expression in bone tissue compared to conventional mice [133]. These investigations consolidate the contribution of gut microorganisms to bone mass homeostasis through modulating host immunity.

Gut bacteria also produce endotoxin LPS, which escalates oxidative stress and inflammation, activating osteoclasts and bone resorption. In osteoblasts, this endotoxin activates NFκB and iNOS (inducible nitric oxide synthase). Gut microorganisms also promote eNOS (endothelial NOS) and increased inflammatory cytokines, including TNF-α, IL-1, and IL-6, repressing trabecular bone mineralization [132]. Estrogen deficiency decreases intestinal permeability and represses estrogen receptor-mediated signaling pathways, such as GTP-binding protein Ras, Raf, IFN-γ (interferon-γ), and MAPK (mitogen-activated protein kinase) in the intestinal epithelial microenvironment [132]. Langan et al. [134] uncover the contribution of plenty of gut microbiota-derived metabolites to inflammation, angiogenesis, and osteoclastogenesis in arthritic joints. Of the metabolites, indole-3-aldehyde and indole-3-acetic acid activate MyD88-mediated NFκB and MAPK pathways in the development of OP.

### 6.3. Gut Microbiota Control of Bone Metabolism

Four microbial phyla in the gut microenvironment, including Firmicutes, Actinobacteria, Bacteroidetes, and Proteobacteria, were associated with host bone quality. In addition, gut microbiota-derived metabolites such as small organic acids, polysaccharides, peptidases, lipids, and choline metabolites are critical for host immune responses and tissue metabolism. The analysis of UPLC-MS/MS (ultra-high-performance liquid chromatography coupled to tandem mass spectrometry) and 16R rRNA sequencing in a cohort study reveals taxonomic metabolome and microbiome in patients with OP [135,136]. The abundances of *Blautia*, *Actinobacillus*, *Oscillospira*, *Bacteroides,* and *Phascolarctobacterium* positively correlate with OP, whereas *Veillonellaceae*, *Collinsella,* and *Ruminococcaceae* negatively correlate with the bone disease. In addition, metabolites from tyrosine and tryptophan metabolism are present in feces and serum of osteoporotic patients along with valine, leucine, and isoleucine degradation. These data highlight the biological significance of gut microbes and their metabolites in the human OP.

## 7. The Biological Contribution of Microbiota-Derived Metabolites to Bone Integrity

Diets directly affect gut microorganism profiles and metabolite production, which influence host tissue function by metabolizing nutrients in diets into small molecules, such as amino acids, fatty acids, organic acid, and vitamins. Of metabolites, bile acid and TMAO, a gut microflora-derived metabolite, correlate with a plethora of metabolic syndrome and tissue degeneration. The contributions of these two molecules to bone tissue warrant a review.

### 7.1. Bile Acid

Bile acid (BAs) is an amphiphilic molecule mainly produced in the liver microenvironment. BAs are critical to the metabolism of cholesterol, phospholipids, and bilirubin and interact with the gastrointestinal tract and liver metabolism. BAs are found to maintain gut microbiota composition and host immune response [137]. The BAs metabolism dysregulation correlates with human bone mass [138]. Supplement with S-propargyl-cysteine, an amino acid analog, promotes *Bifidobacterium* abundance, which is essential to producing bile acid hydrolase to improve bile acid metabolism [139]. *Bifidobacterium* is a source of probiotics that are considered as a probiotic and alleviate inflammatory disorders, including obesity, bowel disease, non-alcoholic fatty liver, and RA. BAs regulate metabolic activity and inflammatory reactions through the TGR5 (G protein-coupled BA receptor) [140]. TGR5 knockout mice show high osteoclast formation, whereas osteoblastic activity is unaffected. Activating TGR5 signaling by agonists changes the AMPK pathway together with intracellular cAMP signaling, repressing osteoclast formation [141].

### 7.2. Trimethylamine N-oxide (TMAO)

It has been more than 100 years since TMAO was first identified, while the investigation confirmed that TMAO concentrations in blood and urine are related to chronic diseases, such as cardiovascular disease [142], atherosclerosis [143], tissue inflammation [144], and cancer development [145]. Increasing evidence has uncovered that TMAO increases oxidative stress and impairs calcium metabolism, mediates the development of age-induced OP [146,147]. Excessive nutrition, obesity, and metabolic disorders will increase the concentration of TMAO in the blood, while the source of circulating TMAO is the precursor TMA (trimethylamine) produced by the intestinal microbiome metabolism of dietary choline. FMO3 (Flavin-containing monooxygenase 3) increases aged tissue and converts TMA into TMAO. Lin et al. [146] reveal that gut dysbiosis and metabolite dysregulation increased serum TMAO levels and hepatic FMO3 levels in aged mice. The decreased abundance of *Bifidobacterium* and *Lactobacillus* together with high abundance of *Clostridium*, *SMB53*, *Oscillospira*, *Sutterella*, *Desulfovibrio,* and *Coprococcus* correlates with serum trimethylamine, TMAO, and oxidative stress in aged mice. Analysis of UHPLC-MS/MS reveals increases in serum TMAO in menopausal women and correlates with bone fracture and OP [148]. High TMAO significantly increases adipocyte formation of bone-marrow stromal cells together with increased oxidative stress and inflammatory cytokine production, including IL-1β, IL-6, and TNF-α. TMAO regulates NF-kB pathways to control osteoblast and adipocyte formation of bone marrow stromal cells [149]. In addition, Li et al.’s in vitro and in vivo assays demonstrated that TMAO is capable of osteogenic differentiation via the ER and mitochondrial stress pathways [150]. Notably, TMAO was able to affect mitochondrial bioenergy metabolism and alterations in mtDNA (mitochondrial DNA) methylation [151], confirming that TMAO is involved in the multifaceted role of pathophysiologies.

## 8. Conclusions

Collective evidence has revealed that gut microbiota correlates with human OA and OP. Gut dysbiosis upregulates inflammation, mitochondrial dysfunction, and oxidative stress by affecting immune responses or metabolite-mediate chondrocyte and bone cell dysfunction, accelerating joint and bone disorders (Figure 1). Supplement with probiotics, microorganism-derived metabolites, or fecal microbiota transplantation delays the development of experimental OA or OP in rodents. More multicenter or double-blind trials are required to verify whether these strategies have remedial effects on human OA or OP. This review sheds productive insights into the biological roles of gut microorganisms to mitochondrial function, inflammation, and redox reactions in host bone-forming cells and chondrocytes during OA and OP, as well as highlights the bone and cartilage-promoting actions of probiotics and metabolites.

## Figures and Tables

**Figure 1 biomedicines-10-00860-f001:**
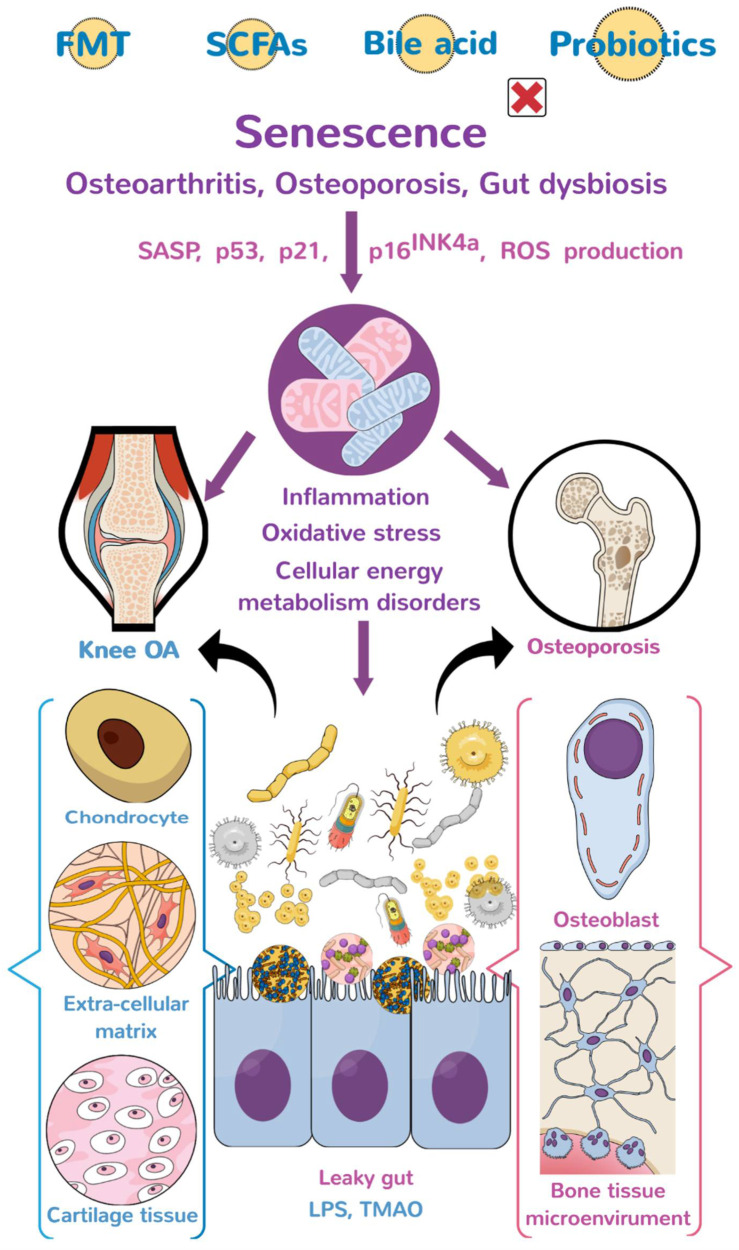
The diagram of the gut-joint axis and gut-bone axis. Gut dysbiosis aggravates inflammation, mitochondrial dysfunction, oxidative stress, and dysregulation of cellular energetics through liposaccharides and metabolites, induced extracellular matrix underproduction, and senescent bone cell (chondrocyte) overburden, which accelerate the development of osteoporosis (OP) and osteoarthritis (OA). The graphs of bone, knee, osteoblasts, chondrocytes, mitochondria, and gut microorganisms are subscribed from Mind the Graph (https://mindthegraph.com, accessed on 1 January 2020), which authorizes the author (W.-S.L.) all the rights to use. FMT: fecal microbiota transplantation; SCFAs: short-chain fatty acids; SASP: senescence-associated secretory phenotype; LPS: lipopolysaccharide; and TMAO: Trimethylamine-N-oxide.

**Table 1 biomedicines-10-00860-t001:** Multifaceted functions of gut microbiota to host tissue metabolism and deterioration.

Host-Microbiota Axis	Predominant Microorganisms	Biological Functions	References
Endothelium-IGF1R	*Akkermansia*	Metabolic homeostasis	[13,14]
Pancreas	*Parabacteroides*, *Escherichia-Shigella*	Development of acute pancreatitis and pathogenesis of acute pancreatitis	[15,16]
Inosine-A_2A_R/PPARγ	*Proteobacteria*, *Enterobacteriaceae*	Maintenance of intestinal homeostasis	[17]
Skin-axis	*Lactobacillus* *plantarum HY7714*	Protection against skin aging	[18]
Myeloid cell-specific type 1 interferon-CCL5	*Debaryomyces hansenii*	Inhibition of inflamed Crohn’s disease and promotion of mucosal healing	[19]
Lung	Coprobacillus, *Clostridium ramosum*, and *Clostridium hathewayi*	Correlation with COVID-19 severity	[20,21]
	*Bacteroides thetaiotaomicron*	Anti-inflammation	
Heart	*Lactobacillus (L) rhamnosus GG*, *Bifidobacterium breve*, *L. casei*, *L. bulgaricus*, and *L. acidophilus*	Activation of NRF-2, preservation of antioxidant, inhibition of NF-κB activity, and reduction of cardiac inflammation	[22]
Brain	*Akkermansia muciniphila*, *Clostridium butyricum*	Reduction of neuroinflammation and improvement of intestinal barrier function in Alzheimer’s disease	[23,24]
	*Lactobacillus brevis*, *Bifidobacterium dentium*, Bacillus (*B. cereus*, *B. mycoides* and *B. subtilis*), Serratia (*S. marcescens*, *S. aureus*), *Proteus vulgaris*, and *Escherichia coli*	Change of neurotransmitter inhibitor—GABA, and neurotransmitter—dopamine production in Parkinson’s and Alzheimer’s disease	[25]
	*Bacteroides ovatus* (*970ed_Bacteroides ovatus*, *054dc_Bacteroides ovatus*), *Parabacteroides merdae* (4ae7e_Parabacteroides) and *Eisenbergiela tayi* (*02b40_Lachnospiraceae*, *29857_Lachnospiraceae*)	Correlation with autism spectrum disorder development	[26]
Liver	*Bifidobacterium*, *Faecalibacterium*, *Oscillospira*, *Ruminococcus*, *Barnesiellaceae*, and *Christensenellaceae*	Regulation of mitochondrial redox and reduction of oxidative stress, bile acid metabolism and reshaping gut microbiota composition; reduction of hepatic steatosis and enhancement of the folate-mediated signaling pathways in mice	[27,28]
Muscle	*Bacillus subtilis C-3102*, *Lactobacillus rhamnosus GG (LGG)*, *Lactobacillus reuteri*, and *Lactobacillus helveticus*	Correlation with muscle strength, and sarcopenic disorders	[29,30]
Bone	*Bacillus subtilis C-3102* *Lactobacillus rhamnosus GG (LGG)*, *Lactobacillus reuteri*, and *Lactobacillus helveticus*	Increases in hip BMD by regulating bone resorption, bone formation, and prevention of diabetes-induced bone loss, and increase in serum calcium levels serum parathyroid hormone (PTH)	[31,32]
Male reproduction	*Prevotella*, *Clostridium scindens*, *Ruminococcus gnavus*, *Butyricicoccus desmolans* and *Clostridium scindens ATCC 35704*; *Clostoridium XVIII*, *Allobaculum*, *Bifidobacterium*, *Eubacterium*, and *Anaerotruncus*	Intestinal flora diversity significantly correlative with sexual hormone activity, such as testosterone, dihydrotestosterone, and androgen. In addition, some major bacterial abundances are associated with erectile dysfunction.	[33,34]
Female reproduction	*Lactobacillales*, *Ruminococcus*, *Clostridium scindens*, *Faecalibacterium*, *Bifidobacterium*, and *Blautia*	Correlation with estrogen circulation concentrations and implicated polycystic ovary syndrome, endometrial hyperplasia, and ultimately fertility.	[35,36]

**Table 2 biomedicines-10-00860-t002:** Characterize therapeutic efficacy of fecal microbiota transplantation in clinical trials.

Human Diseases	Microbiome Enrichment	Post-FMT Efficacy	Study
Sclerosing Cholangitis	*Desulfovibrio* and *Faecalibacterium*; *Odoribacter*, *Alistipes*, and *Erysipelotrichaceae incertae sedis*	Significantly decreased alkaline phosphatase (ALP) and SCFAs producing	[108]
*Clostridioides difficile* infection (CDI)	*family Lachnospiraceae*	Via the metagenomic sequencing analysis of donor and recipient gut microbiome diversity, this information provided positive and successful engraftment, and CDI was cured.	[109]
Immune checkpoint inhibitors (ICI)-associated colitis	*Akkermansia*, *Bifidobacterium* *Blautia*, *Escherichia*, and *Bacteroides*	The study suggests that modulation of the gut microbiome via FMT can significantly improve refractory ICI-associated colitis.	[110]
Severe colitis associated with graft-versus-host disease (GvHD)	*Corynebacterium jeikeium*, *Candida dubliniensis* and *Sporisorium reilianum*	Study showed that bacterial, fungal, and viral communities responded differently to multiple FMTs and understanding the role and importance of reconstituting the gut fungi and viruses.	[111]
anti–PD-1–refractory metastatic melanoma	*Veillonellaceae family*, *Bifidobacterium bifidum*, *Lachnospiraceae*, *Veillonellaceae* and *Ruminococcaceae*	Demonstrated that FMT was associated with favorable changes in immune cell infiltrates in tumor microenvironment and reinduction of anti–PD-1 immunotherapy in 10 patients with anti–PD-1–refractory metastatic melanoma.	[112]

## Data Availability

The data that support the findings of this study are available on request from the corresponding author.

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
