# Peer review of "Gut Microbiota Ecosystem Governance of Host Inflammation, Mitochondrial Respiration and Skeletal Homeostasis"

_biomedicines, 2022, doi:10.3390/biomedicines10040860_

Round 1

Reviewer 1 Report

This review covers the biological significance of gut microorganisms and metabolites for osteoarthritis and osteoporosis. It provides insights into the roles of gut microorganisms to mitochondrial function, inflammation, and reactions in host bone-forming cells and chondrocytes.

  1. Figure 1: Provide the explanations with precise title for this figure. The full names of abbreviations are required.
  2. Table 1: How about the association of reproductive systems with gut microbiota? Most of organs are presented, however, these organs are omitted. 
  3. In addition to cytokines related to inflammation, the review for the correlation with immune check point would be useful. Although the articles for immune check points are rarely reported, they play important roles in inflammation.

Reviewer 2 Report

In this review, the authors want to summarize the potential roles of gut microbiota and their associated metabolites and components in the pathogenesis of osteoporosis (OP) and osteoarthritis (OA) and their treatment. The topic is ok. However, across the manuscript, some contexts were repeated, such as the role of metabolites of gut microbiota in bone function and metabolism (4.3., 5.2.,6.3., section 7). The abbreviations of several molecules are repeatedly shown across the manuscript, such as short-chain fatty acids, MMPs, TLRs, ILs, etc; however, some ones such as PGC-1a were not given the full names.   In addition, Figure 1 without legend was only mentioned in the Conclusion.  The manuscript should be re-organized by the authors to make it more logical. Furthermore, clinical studies should be reviewed to make the topic more interesting.  Some grammar errors should be corrected, such as line 61, the biodiversity that microbiome, line 153,

Round 2

Reviewer 2 Report

Authors made substantial revision for the manuscript. No additional comments.